# Penile Squamous Cell Carcinomas in Sub-Saharan Africa and Europe: Differential Etiopathogenesis

**DOI:** 10.3390/cancers14215284

**Published:** 2022-10-27

**Authors:** Carolina Manzotti, Laurina Chulo, Ricardo López del Campo, Isabel Trias, Marta del Pino, Ofélia Saúde, Iracema Basílio, Nelson Tchamo, Lucilia Lovane, Cesaltina Lorenzoni, Fabiola Fernandes, Adela Saco, Maria Teresa Rodrigo-Calvo, Lorena Marimon, Mamudo R. Ismail, Carla Carrilho, Inmaculada Ribera-Cortada, Jaume Ordi, Natalia Rakislova

**Affiliations:** 1Department of Pathology, Hospital Clínic of Barcelona, University of Barcelona, 08036 Barcelona, Spain; 2Barcelona Institute of Global Health (ISGlobal), University of Barcelona, 08036 Barcelona, Spain; 3Department of Pathology, Maputo Central Hospital, Maputo 1106, Mozambique; 4Clinical Institute of Gynecology, Obstetrics, and Neonatology, Hospital Clínic of Barcelona, University of Barcelona, 08036 Barcelona, Spain; 5Department of Urology, Maputo Central Hospital, Maputo 1106, Mozambique; 6Department of Pathology, Eduardo Mondlane University, Maputo 1106, Mozambique

**Keywords:** penile cancer, Mozambique, Spain, HPV, p16, p53

## Abstract

**Simple Summary:**

We aimed to compare etiopathogenic features of 28 penile squamous cell carcinomas (PSCC) from Mozambique, an African country with a high prevalence of human papillomavirus (HPV) and HIV, with 51 PSCC from Spain, a European country with a low prevalence of HPV and HIV. All tumors underwent HPV testing and p16 and p53 immunohistochemistry. PSCC with either p16 overexpression and/or positivity for HPV were considered HPV-associated. The two sites showed striking etiopathogenic differences. Patients from Mozambique were significantly younger than those from Spain and showed mostly HPV-associated, p53-wild-type PSCC, contrary to older patients from Spain showing mostly HPV-independent, p53-altered tumors. These data may be valuable for primary prevention of PSCC worldwide.

**Abstract:**

Penile squamous cell carcinomas (PSCC) are classified by the World Health Organization into two categories based on their relationship with the human papillomavirus (HPV): HPV-associated and HPV-independent. We compared a cohort of PSCC from Mozambique, a sub-Saharan country in southeast Africa with a high prevalence of HPV and HIV infection, and Spain, a country in southwestern Europe with a low prevalence of HPV and HIV, to study the distribution of the etiopathogenic categories of these tumors in both sites. A total of 79 PSCC were included in the study (28 from Mozambique and 51 from Spain). All cases underwent HPV-DNA polymerase chain reaction (PCR) testing, genotyping, and immunohistochemistry for p16 and p53. Any PSCC showing either p16 overexpression or HPV-DNA in PCR analysis was considered HPV-associated. Overall, 40/79 (50.6%) tumors were classified as HPV-associated and 39 (49.4%) as HPV-independent. The two sites showed marked differences: 25/28 (89.3%) tumors from Mozambique and only 15/51 (29.4%) from Spain were HPV-associated (*p* < 0.001). HPV16 was the most frequent HPV type identified in 64.0% (16/25) of the HPV-associated tumors from Mozambique, and 60.0% (9/15) from Spain (*p* = 0.8). On average, patients from Mozambique were almost two decades younger than those from Spain (mean age 50.9 ± 14.9 and 69.2 ± 13.3, respectively [*p* < 0.001]). In conclusion, significant etiopathogenic differences between PSCC in Mozambique and Spain were observed, with a remarkably high prevalence of HPV-associated tumors in Mozambique and a relatively low prevalence in Spain. These data may have important consequences for primary prevention of PSCC worldwide.

## 1. Introduction

Penile cancer is an uncommon malignancy with an estimated age-standardized incidence worldwide of 0.80 per 100,000 person-year in 2018 [1] that typically affects men in their fifth to seventh decades of life. Over 95% of penile cancers are penile squamous cell carcinomas (PSCC) [2]. Several risk factors for PSCC have been identified, including sexual behavior, history of warts/condylomas/human papillomavirus (HPV) infection, lack of circumcision, phimosis, poor penile hygiene, smoking and chronic inflammatory conditions such as lichen sclerosus [3]. In its 2016 classification, the World Health Organization (WHO) recommended separating PSCC based on their HPV status into HPV-associated and HPV-independent tumors [4,5], with this categorization having been maintained in the recent WHO 2022 revision [6]. This WHO classification recognizes a clear association between histological types and their relationship with HPV, with several histological variants of PSCC that are typically HPV-associated and other histological variants known as HPV-independent [6].

HPV-associated PSCC frequently arise on an intraepithelial precursor named HPV-associated penile intraepithelial neoplasia (PeIN), a lesion with a basaloid, warty, or warty–basaloid morphology [7], also called high-grade squamous intraepithelial lesion (HSIL) according to the consensus of the Lower Anogenital Squamous Terminology standardization project [8]. HPV-associated PeIN is characterized by immature basal-appearing cells involving the whole thickness of the epithelium. Both HPV-associated PSCC and PeIN show strong, block type staining for p16 immunohistochemistry (IHC) [7,9,10,11]. Contrarily, HPV-independent PSCC frequently arise on a precursor lesion named non-HPV-related PeIN or differentiated PeIN, a lesion characterized by atypical basal cells with normal maturation in the superficial layers that usually develops in a context of inflammatory conditions of the penis, such as lichen sclerosus [7,9,10]. Both HPV-independent PSCC and its precursor lesion show no overexpression of p16 and frequently display an abnormal pattern of p53 [7,9,10].

The incidence of PSCC has marked geographical variability, with the highest incidence centered in low- and middle-income countries (LMIC), especially in southern Africa, South Asia, and South America, and a significantly lower, albeit rising, incidence in most European countries [12]. Unfortunately, while the epidemiology of PSCC in high-income areas is relatively well known, data on PSCC in LMIC, particularly in sub-Saharan Africa, are very sparse. The largest published study encompassing 1010 PSCC, involved only 19 tumors (1.8%) from sub-Saharan African countries [13]. The high prevalence of HPV infection may have a significant impact in terms of the proportion of HPV-associated and HPV-independent tumors. In addition, the high prevalence of HIV in these areas [14] may have also a significant impact in the rates of HPV-associated and -independent PSCC, as this condition multiplies the risk of persistent HPV infection and, consequently, leads to increased incidence of HPV-associated premalignant lesions and carcinomas [15].

The aim of this study was to compare the differential etiopathogenic characteristics of PSCC in Mozambique, a sub-Saharan country in South-East Africa with high prevalence of HPV and HIV and Spain, a European country with low ratesof HPV and HIV [16].

## 2. Methods

### 2.1. Case Selection

The computer records of the Department of Pathology of the Maputo Central Hospital (MCH) and the Hospital Clinic of Barcelona (HCB)-two tertiary referral health institutions in Mozambique and Spain, respectively, were revised looking for all the cases diagnosed with PSCC. The period revised in the MCH was from January 2018 to December 2020, being from January 2000 to December 2020 in the HCB.

All PSCC identified during these periods were retrieved, and the material available was histologically reviewed. The inclusion criteria for the study were: (1) the presence of invasive PSCC and (2) the material available for HPV detection by polymerase chain reaction (PCR), and p16 and p53 IHC.

The ethics committees of both the Faculty of Medicine of the Eduardo Mondlane University & MCH and HCB approved the study (refs. CIBS FM&HCM/071/2017 and HCB/2020/1207, respectively).

### 2.2. Histological Review

Sections of all the tumors stained with hematoxylin and eosin were reviewed by two pathologists (CM, NR) to confirm the presence of invasive carcinoma. Following the WHO criteria, several histological types were considered as probably related to HPV (basaloid, papillary–basaloid, warty, wart–basaloid, clear cell, and lymphoepithelioma-like) and other histological variants were considered as not related to HPV infection (usual type [keratinizing], verrucous [including cuniculatum], papillary and sarcomatoid) [4,6]. After revision, the most representative and well-preserved paraffin block, containing the most obvious pathological findings of each case, was selected for HPV detection and IHC staining.

### 2.3. Tissue Preparation, Nucleic-Acid Isolation and HPV-DNA Detection

DNA extraction was performed on 10µm whole sections of formalin-fixed paraffin-embedded tissue from surgical specimens or penile biopsies. The microtome blade was replaced between cases. No microdissection was performed but, in all cases, the tissue analyzed included the invasive tumor. The highest safety measures were carried out while sectioning and performing sample preparation to avoid cross-contamination. DNA was extracted after overnight incubation in 20 μL of proteinase K solution at 56 °C. Subsequently, the DNA was isolated using a commercial kit (QIAamp DNA FFPE Tissue Kit; Qiagen, Hilden, Germany) according to the manufacturing instructions. Samples were quantified using the NanoDrop 2000 (Thermo Fisher Scientific, Waltham, MA, USA) to determine DNA concentrations for the genotyping test. Samples with less than 10 µL of isolated DNA were excluded from the study.

The SPF10 PCR and LiPA25 system were used for HPV-DNA detection and typing (version 1, Labo Biomedical Products, Rijswijk, The Netherlands). A volume of 10 µL of isolated DNA was PCR-amplified using the INNO-LiPA HPV Genotyping Extra II kit (Fujirebio, Gent, Belgium). The same kit was used for HPV genotyping. This system allows the genotyping of HPV 16, 18, 31, 33, 35, 39, 45, 51, 52, 56, 58, 59, 68, 26, 53, 66, 70, 73, 82, 6, 11, 40, 42, 43, 44, 54, 61, 62, 67, 81, 83 and 89. HPV DNA-positive samples not hybridizing with any of the 32 probes were classified as HPV type X (HPV X or undetermined type).Each run was performed with positive and negative controls to monitor the efficiency of DNA isolation, PCR amplification, hybridization, and genotyping procedures.

### 2.4. p16 IHC

All tumors were stained with a p16 monoclonal antibody using the CINtec Histology Kit (clone E6H4; Roche-mtm-Laboratories, Heidelberg, Germany). The tumors were considered as p16 positive if they showed strong and diffuse block-like staining, and were considered as negative when the staining was patchy or completely negative.

All IHC stains were conducted at the department of Pathology of the HCB, employing the BenchMark ULTRA platform (Roche-Ventana, Tucson, AZ, USA).

### 2.5. Criteria for Classifying a Tumor as HPV-Associated or HPV-Independent

In this study any PSCC showing either p16 IHC overexpression and/or HPV-DNA in the PCR analysis were considered as HPV-associated. All PSCC showing a negative result for the two techniques (p16 IHC and HPV-DNA PCR) were considered as HPV-independent.

As it has recently been proposed that a second biomarker, such as p16 or E6*I mRNA, should also be used to consider a tumor as etiologically associated with HPV [13], a second, more restrictive scenario was also calculated, including only tumors showing both HPV-DNA in the PCR analysis and p16 IHC overexpression as HPV-associated. In this restrictive scenario, tumors showing a discrepant result (HPV-DNA negative and p16 positive or HPV-DNA positive and p16 negative) were considered as HPV-independent.

### 2.6. p53 IHC

A monoclonal antibody was used for p53 staining (clone DO-7, Dako, Carpinteria, CA, USA). The interpretation of staining was conducted in the invasive tumor following the pattern-based framework recently developed for vulvar carcinomas [17,18]. This framework includes two normal (wild-type) and four abnormal (suggestive of mutant protein) patterns. Normal patterns included (a) scattered basal nuclear positivity (b) staining in cells in the mid-epithelial layers, with absent basal staining. Abnormal p53 patterns comprised: (1) basal overexpression (strong and continuous staining of the basal cells), (2) diffuse overexpression (strong and continuous staining of the basal and suprabasal cells (3) cytoplasmic staining (with or without nuclear positivity), and (4) null pattern (absence of staining in the tumoral cells with conserved staining in the normal skin, stromal or inflammatory cells).

### 2.7. Statistical Analysis

The SPSS 28.0 statistical package (SPSS, Chicago, IL, USA) was used to perform the data analyses. Categorical variables are shown as absolute numbers and percentages. And were analyzed with the chi-square or Fisher exact test. Quantitative variables are shown as mean ± standard deviation, and were compared with Student’s t test or analysis of variance test.

## 3. Results

### 3.1. General Features

Overall, 79 tumors were included in the study, 28 from the MCH (Mozambique) and 51 from the HCB (Spain). The age, HIV status, histological variant, p16 IHC, HPV-DNA PCR, and p53 IHC results of the PSCC from Mozambique and Spain are shown in Table 1 and Table 2, respectively.

The clinical presentation of PSCC was markedly different in the two sites, with the patients from Mozambique being on average almost two decades younger than patients from Spain (mean age of the patients from Mozambique 50.9 ± 14.9 years and mean age in Spain 69.2 ± 13.3 years [*p* < 0.001]). HIV status was available in 8 cases from Mozambique and in all cases from Spain. Six out of 8 (75.0%) of the Mozambican patients were HIV positive in contrast with 1/51 (1.9%) of the Spanish patients.

### 3.2. Association with HPV

Overall 40/79 (50.6%) tumors were classified as HPV-associated and 39 (49.4%) as HPV-independent. However, marked differences were observed between the two sites: 25/28 (89.3%) tumors from Mozambique vs. only 15/51 (29.4%) from Spain were HPV-associated. Using stricter criteria (restrictive scenario, i.e., HPV-DNA identified by PCR and p16 IHC overexpression) 20/28 PSCC (71.4%) from Mozambique and 9/51 (17.6%) from Spain were considered as HPV-associated. In both scenarios the differences were statistically significant (*p* < 0.001).

HPV16 was the most frequent HPV type identified in 64.0% (16/25) of the HPV-associated tumors from Mozambique and in 60.0% (9/15) of the tumors from Spain (*p* = 0.8). No differences in the distribution of HPV type were identified between the two sites in any of the scenarios (*p* = 1). The prevalence of infection by multiple HPV types was slightly higher in Mozambique than in Spain, although the differences did not reach statistical significance (28.0% [7/25] vs. 6.7% [1/15], respectively; *p* = 0.219).

### 3.3. Histological Subtypes

Of the 40 tumors associated with infection by HPV, 36 (90.0%) were of the histological variants defined as related to HPV according to the WHO (16 [40.0%] basaloid, 7 [17.5%] warty, 12 [30.0%] mixed basaloid/warty, and 1 [2.5%] lymphoepithelioma-like), but four (10.0%) were of the usual type, one of the variants considered as not related to HPV by the WHO. Contrarily, 33/39 (84.6%) of the HPV-independent PSCC were of the histological variants defined as not related to HPV according to the WHO (29 [74.4%] usual type, 3 [7.6%] verrucous, 1 [2.4%] cuniculatum), but 6 (15.4%) were variants classified as HPV-related by the WHO (2 [5.1%] basaloid, 2 [5.1%] warty, and 2 [5.1%] mixed warty/basaloid).

### 3.4. p53 IHC Results

p53 IHC showed a normal staining pattern suggestive of wild-type protein in 35/40 (87.5%) of the HPV-associated tumors, and only 5/40 (12.5%) showed an abnormal pattern suggestive of a mutated p53. No differences were observed between Mozambique and Spain (*p* = 0.38). Contrarily, the HPV-independent PSCC showed an abnormal pattern of p53 IHC staining in 24/39 (61.5%) and a normal pattern in 15/39 (38.5%) (*p* < 0.0001).

Figure 1 shows an example of the histological and IHC features of the HPV-associated and HPV-independent types of PSCC.

## 4. Discussion

In this study we provide relevant information in relation to the epidemiology of PSCC in sub-Saharan Africa, a geographical area in which these data are particularly scant. Our series of PSCC from sub-Saharan Africa and Europe shows relevant etiopathogenic differences between the two sites and highlights the extremely high prevalence of HPV-associated PSCC in Mozambique compared with the low prevalence in Spain. Over 89% of the PSCC in Mozambique but only 29% of the PSCC from Spain were HPV-associated (*p* < 0.001). Using the restrictive scenario (i.e., p16 IHC overexpression and HPV-DNA identified by PCR), the proportion of HPV-associated tumors from Mozambique was still extremely high (71.4%), whereas the percentage of HPV-associated tumors in Spain dropped to 17.6% (*p* < 0.001). HPV16 was the most frequent HPV type identified in both sites (64.0% and 60.0% HPV-associated tumors from Mozambique and Spain, respectively), similar to what has been reported in other studies [13,19].

The results of our Spanish cohort are in keeping with most studies from high-income countries. In these sites, PSCC is typically a disease of old men, and a significant proportion arise independently from HPV, frequently within the background of inflammatory conditions of the penile skin, such as lichen sclerosus and lichen simplex chronicus [13,14,20]. Contrarily, information from LMIC sites is very scant [21], but existing data suggest that PSCC may have a distinctive epidemiological background, with most tumors arising via an HPV-associated pathway [22]. A high prevalence of HPV infections has been reported in many LMIC, with Mozambique showing an incidence as high as 63% [23]. As HPV-associated tumors tend to affect younger individuals, patients from Spain were almost two decades older in average than patients from Mozambique, a pattern also observed in vulvar squamous cell carcinomas [24].

Although the high prevalence of HPV infection should be considered the main factor the high prevalence of HPV-associated PSCC in the Mozambican cohort, the high prevalence of HIV in Mozambique [25] has been suggested as a relevant contributor to the rise in HPV-associated carcinomas in Mozambique and also in other sub-Saharan countries [26]. A community-based study in a rural district of Maputo Province (Mozambique) showed a 37% prevalence of HIV in men 28–47 years old [25]. A high risk of developing HPV-related cancers [27] has been observed in people living with HIV, and the two viruses (HPV and HIV type 1) have been categorized as carcinogens by the International Agency for Research on Cancer [28]. Nevertheless, whereas the association between HIV and cervical cancer has been well established [29], the link between HPV-associated PSCC and HIV has not been evaluated. Interestingly, 75% of the Mozambican patients for whom clinical data were available were HIV positive. In contrast, the prevalence of HIV-positive patients in the Spanish series was 1.9%. Finally, the limited access to the health system might also contribute to this high incidence, as a large part of population in Mozambique lives in rural areas where access to health care is poor. 

Interestingly, although 90% of the HPV-associated tumors were basaloid, warty, mixed basaloid/warty, or lymphoepithelioma-like, the types of PSCC considered, following the WHO classification, as tumors related to HPV [4,6], 10% of the HPV-associated PSCC in our series were of the usual type. Similarly, whereas 85% of the HPV-independent tumors were of the usual type or verrucous, 15% were basaloid, warty, or mixed warty/basaloid, in other words, the histological types considered in this classification as related to HPV. Similar results have been reported in the largest study on penile tumors [13]. These findings suggest that diagnosis based on pure morphological criteria may be misleading in a small but significant proportion of tumors and indicate that p16 IHC and/or HPV-DNA testing should be required to properly classify PSCC as HPV-associated or HPV-independent. In this regard, we recently reported that a small proportion of HPV-independent intraepithelial precursors of penile cancer may be indistinguishable from the HPV-associated precursors [30]. This overlap between HPV-associated and -independent tumors is a well-known phenomenon in vulvar tumors [31] and head and neck tumors, and the WHO classification of both types of neoplasms stresses the need for p16 IHC and/or HPV-DNA testing to adequately classify these tumors because of the better prognosis of HPV-associated compared with HPV-independent tumors in all these sites [32,33]. Growing evidence suggesting that this better behavior of HPV-associated tumors also applies to PSCC [34,35] stresses the need for these ancillary studies to properly classify PSCC. Nevertheless, ancillary studies to detect HPV in the tumor are probably not necessary in sub-Saharan countries, as most PSCC are HPV-associated given that HPV-independent tumors are extremely infrequent and laboratory capacities are low [36].

Interestingly, abnormal p53 IHC-mutant patterns were identified in a high percentage of HPV-independent PSCC and in only a small percentage of HPV-associated tumors, a phenomenon already reported in studies based on IHC, in addition to studies analyzing TP53 mutations [37,38,39].

A subset of tumors showed discrepant p16 IHC and HPV-DNA results. It is difficult to draw conclusions on the significance of these findings and on whether these cases are or are not etiologically related to HPV with the data of the present study. It has recently been proposed that in addition to the identification of the DNA of the HPV, a second biomarker, such as p16 or E6 mRNA, should also be required to consider a tumor as etiologically associated with HPV [13]. On the other hand, studies conducted in vulvar squamous cell carcinoma suggest that p16 IHC is more reliable than HPV-DNA detection tests using formalin-fixed, paraffin-embedded material [31].

## 5. Conclusions

Most PSCC in Mozambique, a country in southeastern Africa, which probably reflects the epidemiological picture of many sub-Saharan African countries, are HPV-associated and arise in young men. In contrast, most PSCC diagnosed in Europe are HPV-independent and affect old men. These data suggest that extending HPV vaccination programs to males in LMIC could, in addition to the benefits of tackling the cervical cancer burden, reduce a significant proportion of penile cancer.

## Figures and Tables

**Figure 1 cancers-14-05284-f001:**
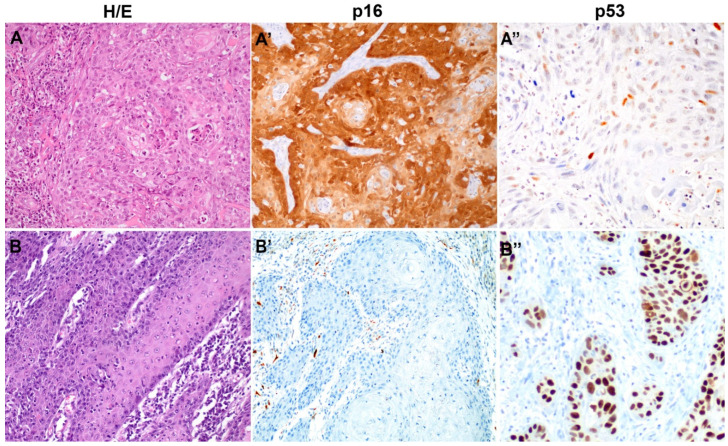
Histological and immunohistochemical features of a representative example of Human Papillomavirus (HPV)-associated (**A**,**A′**,**A″**) and HPV-independent (**B**,**B′**,**B″**) penile squamous cell carcinoma (PSCC). (**A**,**B**, hematoxylin and eosin, 200×), p16 (**A′**,**B′**) and p53 (**A″**,**B″**), HPV-associated PSCC, the most frequent type in Mozambique, is characterized by diffuse positive p16 staining and scattered irregular nuclear positivity for p53. HPV-independent PSCC, the most frequent type in Spain, is characterized by keratinizing histology, negative p16 staining and marked overexpression of p53.

**Table 1 cancers-14-05284-t001:** Age, HIV status, histological variant, Human Papillomavirus (HPV)-DNA result, p16 and p53 immunohistochemical features of the tumors from Mozambique.

Case	Age	HIV Status	Histological Variant	p16 IHC	HPV-DNA Detection and Type	p53 IHC
**HPV-associated carcinomas**						
M2	40	NA	Warty/basaloid	+	70	Normal
M3	40	+	Usual type	+	45, 51, 70	Normal
M5	98	NA	Warty/basaloid	+	16	Normal
M7	44	NA	Basaloid	+	16	Normal
M8	38	NA	Warty/basaloid	+	18, 39	Normal
M10	62	−	Warty	+	16	Normal
M11	38	NA	Warty/basaloid	+	16	Normal
M12	39	NA	Warty/basaloid	+	16	Normal
M14	51	NA	Warty	+	18, 31, 44, 6	Normal
M15	41	NA	Warty/basaloid	+	16	Normal
M16	34	NA	Warty/basaloid	+	16	Normal
M17	30	NA	Basaloid	+	16	Normal
M18	39	NA	Warty/basaloid	+	16	Normal
M20	43	+	Basaloid	+	16	Abnormal
M22	66	NA	Basaloid	+	68	Normal
M23	45	+	Warty/basaloid	+	16, 68	Normal
M24	51	+	Basaloid	+	16	Normal
M25	43	+	Warty	+	16	Normal
M27	68	+	Basaloid	+	16, 52, 82, 11	Normal
M28	52	−	Basaloid	+	16	Normal
M6	56	NA	Warty	+	Negative	Normal
M26	59	NA	Basaloid	+	Negative	Normal
M1	68	NA	Warty/basaloid	−	16, 11	Abnormal
M9	51	NA	Usual type	−	X	Normal
M13	53	NA	Basaloid	−	18, 31, 52	Normal
**HPV-independent carcinomas**						
M4	65	NA	Warty	−	Negative	Abnormal
M19	74	NA	Warty	−	Negative	Abnormal
M21	38	NA	Usual type	−	Negative	Abnormal

NA: not available: + positive; − negative; X: undetermined HPV genotype.

**Table 2 cancers-14-05284-t002:** Age, HIV status, histological variant, Human Papillomavirus (HPV)-DNA result, p16 and p53 immunohistochemical features of the tumors from Spain.

Case	Age	HIV Status	Histological Variant	p16	HPV-DNA Detection and Type	p53
**HPV-associated carcinomas**						
S3	51	−	Basaloid	+	16	Abnormal
S9	61	−	Warty	+	16	Normal
S14	58	−	Basaloid	+	16	Normal
S21	52	−	Usual type	+	16, 18	Normal
S22	81	−	Lymphoepithelioma-like	+	16	Normal
S38	45	+	Warty-basaloid	+	16	Normal
S45	67	−	Warty	+	6	Normal
S46	87	−	Warty-basaloid	+	16	Normal
S51	59	−	Basaloid	+	16	Normal
S12	60	−	Warty	+	−	Abnormal
S29	94	−	Basaloid	+	−	Normal
S30	63	−	Basaloid	+	−	Normal
S31	64	−	Basaloid	+	−	Normal
S13	60	−	Usual type	−	16	Abnormal
S49	76	−	Basaloid	−	X	Normal
**HPV-independent carcinomas**						
S1	62	−	Verrucous	−	−	Normal
S2	65	−	Usual type	−	−	Normal
S4	59	−	Usual type	−	−	Abnormal
S5	53	−	Usual type	−	−	Abnormal
S6	53	−	Usual type	−	−	Abnormal
S7	78	−	Usual type	−	−	Abnormal
S8	54	−	Usual type	−	−	Abnormal
S10	81	−	Usual type	−	−	Abnormal
S11	83	−	Usual type	−	−	Normal
S15	74	−	Basaloid	−	−	Abnormal
S16	70	−	Usual type	−	−	Abnormal
S17	81	−	Usual type	−	−	Abnormal
S18	85	−	Warty-basaloid	−	−	Normal
S19	63	−	Usual type	−	−	Normal
S20	59	−	Usual type	−	−	Abnormal
S23	80	−	Usual type	−	−	Normal
S24	72	−	Basaloid	−	−	Abnormal
S25	74	−	Usual type	−	−	Normal
S26	73	−	Usual type	−	−	Normal
S27	77	−	Usual type	−	−	Abnormal
S28	65	−	Usual type	−	−	Abnormal
S32	85	−	Verrucous	−	−	Normal
S33	79	−	Usual type	−	−	Abnormal
S34	75	−	Usual type	−	−	Normal
S35	86	−	Usual type	−	−	Abnormal
S36	70	−	Verrucous (cuniculatum)	−	−	Normal
S37	56	−	Usual type	−	−	Abnormal
S39	59	−	Usual type	−	−	Abnormal
S40	40	−	Usual type	−	−	Abnormal
S41	83	−	Verrucous	−	−	Normal
S42	75	−	Usual type	−	−	Abnormal
S43	44	−	Usual type	−	−	Abnormal
S44	67	−	Warty-basaloid	−	−	Abnormal
S47	96	−	Usual type	−	−	Normal
S48	85	−	Usual type	−	−	Normal
S50	89	−	Usual type	−	−	Normal

NA: not available: + positive; − negative; X: undetermined HPV genotype.

## Data Availability

All the data is shown in the manuscript.

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
