# Peer review of "Penile Squamous Cell Carcinomas in Sub-Saharan Africa and Europe: Differential Etiopathogenesis"

_cancers, 2022, doi:10.3390/cancers14215284_

Round 1

Reviewer 1 Report

General comments

In the present study, the author compared between etiopathogenic features of penile squamous cell carcinomas (PSCC) from Mozambique with a high prevalence of human papillomavirus (HPV) and HIV, with PSCC from Spain with a low prevalence of HPV and HIV. It is very interesting to note that HPV-associated PSCC is more common in Mozambique, while HPV-independent PSCC is more common in Spain. The implications of this paper are important because different etiologies have different prognoses and preventive measures. This study may contribute to informing the importance of the HPV vaccine for men, especially in low and middle income countries. There are a few issues which should be addressed:

Specific comments for revision:

1.     Although the rate of HPV-associated PSCC has been shown to be lower in Europe than in Africa, the incidence of HPV-associated PSCC in Spain in this study appears to be lower than in previous reports. This may be due to the strict criteria for HPV-related PSCC (HPV detection and p16 overexpression). If possible, please add to the discussion the rates of HPV-associated PSCC in Europe and Africa in previous reports. In addition, it is not impossible that a false negative result could have been obtained due to insufficient DNA copy number, so the method of DNA extraction needs to be described in more detail, such as whether it was confirmed that sufficient DNA extraction was performed.

2.     Although not everyone in Mozambique is able to test for HIV, it would be better to add a section on HIV status to Table 1 as well as Table 2. Also, please unify the titles of Table 1 and 2 regarding HPV-DNA, as they are different (HPV-DNA and HPV-DNA detection and type). Does X in the table refer to NA? Please correct if necessary.

Author Response

Please find attached the point-by-point response to the reviewer's comments

Reviewer 2 Report

The manuscript is about regional differences investigated the prevalence of HPV-dependent penile cancer. They used the correct pathological and molecular cytological method with a large sample size. However, the following components are lacking:

 Method: L111-112- need a more detailed description of “the most representative” paraffin block used in the study. For example, “more obvious pathological findings”, etc. As you know, it is common to find areas where HPV-DNA is extracted and cites where it is not, even within the same tumor tissue, which could affect the results of this study.

 L250-261: The author has considered the high incidence of HPV-associated cancers in relation to the prevalence of HIV. However, this alone is not sufficient to explain the high incidence of HPV-associated cancers. More convincing arguments need to be added.

Author Response

Please find attached point-by-point responses to the reviewer's comments. 
